# SQA3D:
# SITUATED QUESTION ANSWERING IN 3D SCENES

**Xiaojian Ma**[2*] , **Silong Yong**[1,3*], **Zilong Zheng**[1] , **Qing Li**[1] , **Yitao Liang**[1,4]
**Song-Chun Zhu**[1,2,3,4] , **Siyuan Huang**[1]
[1]Beijing Institute for General Artificial Intelligence (BIGAI)  [2]UCLA  [3]Tsinghua University
[4]Peking University
xiaojian.ma@ucla.edu, yongzl19@mails.tsinghua.edu.cn
{zlzheng,liqing,sczhu,syhuang}@bigai.ai, yitaol@pku.edu.cn

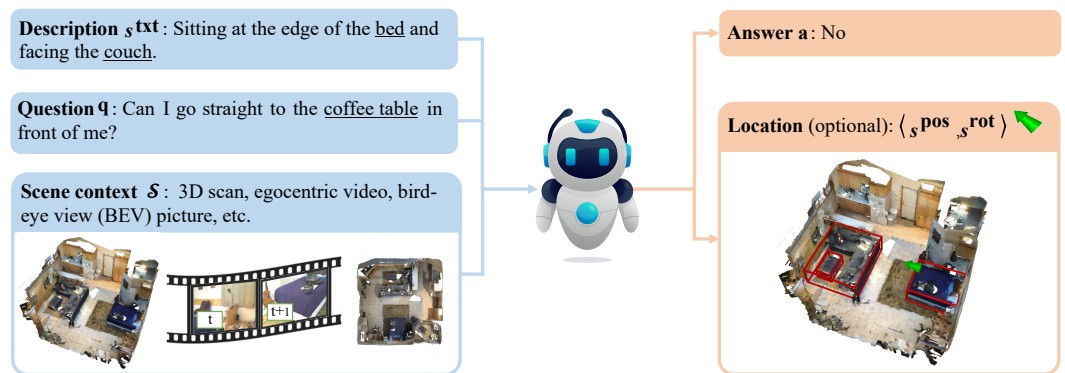

Figure 1: Task illustration of Situated Question Answering in 3D Scenes (SQA3D). Given scene context $\mathcal{S}$ (*e.g.*, 3D scan, egocentric video, bird-eye view picture), SQA3D requires an agent to first comprehend and localize its **situation** (position, orientation, *etc.*) in the 3D scene from a textual description $s^{\text{txt}}$, then answer a question $q$ under that situation. **Note that understanding the situation and imagining the corresponding egocentric view correctly is necessary to accomplish our task.** We provide more example questions in Figure 2.

## ABSTRACT

We propose a new task to benchmark scene understanding of embodied agents: Situated Question Answering in 3D Scenes (SQA3D). Given a scene context (*e.g.*, 3D scan), SQA3D requires the tested agent to first understand its **situation** (position, orientation, etc.) in the 3D scene as described by text, then reason about its surrounding environment and answer a question under that situation. Based upon 650 scenes from ScanNet, we provide a dataset centered around 6.8k unique situations, along with 20.4k descriptions and 33.4k diverse reasoning questions for these situations. These questions examine a wide spectrum of reasoning capabilities for an intelligent agent, ranging from spatial relation comprehension to commonsense understanding, navigation, and multi-hop reasoning. SQA3D imposes a significant challenge to current multi-modal especially 3D reasoning models. We evaluate various state-of-the-art approaches and find that the best one only achieves an overall score of 47.20%, while amateur human participants can reach 90.06%. We believe SQA3D could facilitate future embodied AI research with stronger situation understanding and reasoning capabilities. Code and data are released at sqa3d.github.io.

## 1 INTRODUCTION

In recent years, the endeavor of building intelligent embodied agents has delivered fruitful achievements. Robots now can navigate (Anderson et al., 2018) and manipulate objects (Liang et al., 2019; Savva et al., 2019; Shridhar et al., 2022; Ahn et al., 2022) following natural language commands

---

*First two authors contributed equally. Correspondence to Zilong Zheng and Siyuan Huang.

Figure 2: **Examples from SQA3D.** We provide some example questions and the corresponding situations ($s^{\text{txt}}$ and ↘) and 3D scenes. The categories listed here do not mean to be exhaustive and a question could fall into multiple categories. The green boxes indicate relevant objects in situation description $s^{\text{txt}}$ while red boxes are for the questions $q$.

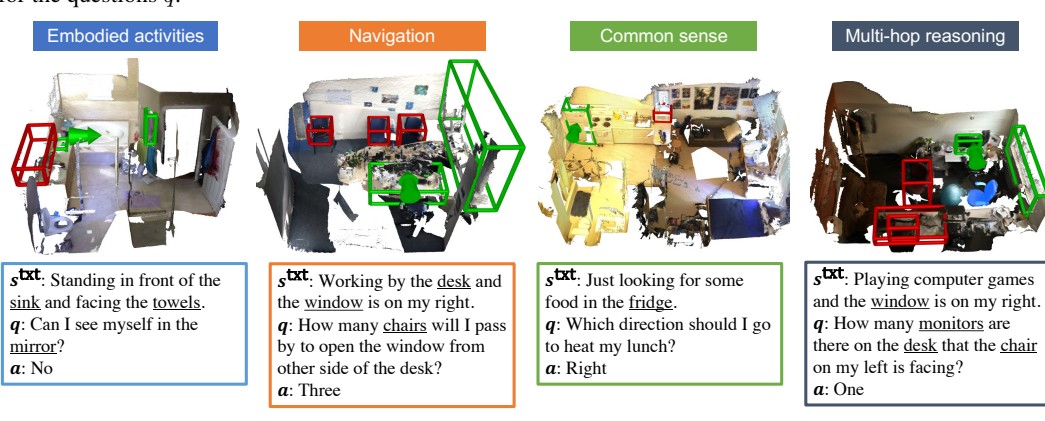

| Embodied activities | Navigation | Common sense | Multi-hop reasoning |
|---|---|---|---|
| $s^{\text{txt}}$: Standing in front of the sink and facing the towels. $q$: Can I see myself in the mirror? $a$: No | $s^{\text{txt}}$: Working by the desk and the window is on my right. $q$: How many chairs will I pass by to open the window from other side of the desk? $a$: Three | $s^{\text{txt}}$: Just looking for some food in the fridge. $q$: Which direction should I go to heat my lunch? $a$: Right | $s^{\text{txt}}$: Playing computer games and the window is on my right. $q$: How many monitors are there on the desk that the chair on my left is facing? $a$: One |

or dialogues. Albeit these promising advances, their actual performances in real-world embodied environments could still fall short of human expectations, especially in generalization to different situations (scenes and locations) and tasks that require substantial, knowledge-intensive reasoning. To diagnose the fundamental capability of realistic embodied agents, we investigate the problem of **embodied scene understanding**, where the agent needs to understand its situation and the surroundings in the environment from a *dynamic* egocentric view, then perceive, reason, and act accordingly, to accomplish complex tasks.

**What is at the core of embodied scene understanding?** Drawing inspirations from situated cognition (Greeno, 1998; Anderson et al., 2000), a seminal theory of embodiment, we anticipate it to be two-fold:

- **Situation understanding.** The ability to imagine what the agent will see from arbitrary situations (position, orientations, *etc*.) in a 3D scene and understand the surroundings anchored to the situation, therefore generalize to novel positions or scenes;
- **Situated reasoning.** The ability to acquire knowledge about the environment based on the agents' current situation and reason with the knowledge, therefore further facilitates accomplishing complex action planning tasks.

To step towards embodied scene understanding, we introduce **SQA3D**, a new task that reconciles the best of both parties, situation understanding, and situated reasoning, into embodied 3D scene understanding. Figure 1 sketches our task: given a 3D scene context (*e.g*., 3D scan, ego-centric video, or bird-eye view (BEV) picture), the agent in the 3D scene needs to first comprehend and localize its situation (position, orientation, *etc*.) from a textual description, then answer a question that requires substantial situated reasoning from that perspective. We crowd-sourced the situation descriptions from Amazon MTurk (AMT), where participants are instructed to select diverse locations and orientations in 3D scenes. To systematically examine the agent's ability in situated reasoning, we collect questions that cover a wide spectrum of knowledge, ranging from spatial relations to navigation, common sense reasoning, and multi-hop reasoning. In total, SQA3D comprises 20.4k descriptions of 6.8k unique situations collected from 650 ScanNet scenes and 33.4k questions about these situations. Examples of SQA3D can be found Figure 2.

Our task closely connects to the recent efforts on 3D language grounding (Dai et al., 2017; Chen et al., 2020; 2021; Hong et al., 2021b; Achlioptas et al., 2020; Wang et al., 2022; Azuma et al., 2022). However, most of these avenues assume observations of a 3D scene are made from some third-person perspectives rather than an embodied, egocentric view, and they primarily inspect *spatial understanding*, while SQA3D examines scene understanding with a wide range of knowledge, and the problems have to be solved using an (imagined) first-person view. Embodied QA (Das et al., 2018; Wijmans et al., 2019a) draws very similar motivation as SQA3D, but our task adopts a simplified protocol (QA only) while still preserving the function of benchmarking embodied scene understanding, therefore allowing more complex, knowledge-intensive questions and a much larger scale of data collection. Comparisons with relevant tasks and benchmarks are listed in Table 1.

**Benchmarking existing baselines**: In our experiments, we examine state-of-the-art multi-modal reasoning models, including ScanQA from Azuma et al. (2022) that leverages 3D scan data, Clip-

Table 1: **An overview of the different benchmark datasets covering grounded 3D scene understanding.** In general, we consider semantic grounding, language-driven navigation, and question-answering in photo-realistic 3D scenes. In the first row, *situated* indicates whether the benchmark task is supposed to be completed by a "situated" agent with its egocentric perspective. *navigation*, *common sense*, and *multi-hop reasoning* show whether the task requires a certain capability or knowledge level of 3D understanding. *Rather than observing a complete 3D scan of the scene, the learner needs to navigate in a simulator to perceive the 3D scene incrementally.

| dataset | task | situated? | 3D type | text collection | navi-gation? | common sense? | multi-hop reasoning? | #scenes | #tasks |
|---|---|---|---|---|---|---|---|---|---|
| ScanNet (Dai et al., 2017) | seg. | ✗ | scan | n/a | ✗ | ✗ | ✗ | 800 rooms | 1.5k |
| ScanRefer (Chen et al., 2020) | det. | ✗ | scan | human | ✗ | ✗ | ✗ | 800 rooms | 52k |
| ReferIt3D (Achlioptas et al., 2020) | det. | ✗ | scan | human | ✗ | ✗ | ✗ | 707 rooms | 41k |
| ScanQA (Azuma et al., 2022) | q.a. | ✗ | scan | template | ✗ | ✗ | ✗ | 800 rooms | 41k |
| 3D-QA (Ye et al., 2021) | q.a. | ✗ | scan | human | ✗ | ✗ | ✗ | 806 rooms | 5.8k |
| CLEVR3D (Yan et al., 2021) | q.a. | ✗ | scan | template | ✗ | ✗ | ✓ | 478 rooms | 60k |
| MP3D-R2R (Anderson et al., 2018) | nav. | ✓ | *nav. | human | ✓ | ✗ | ✗ | 190 floors | 22k |
| MP3D-EQA (Wijmans et al., 2019a) | q.a. | ✓ | *nav. | template | ✓ | ✗ | ✗ | 146 floors | 1.1k |
| SQA3D (Ours) | q.a. | ✓ | scan | human | ✓ | ✓ | ✓ | 650 rooms | 33.4k |

BERT (Lei et al., 2021) and MCAN (Yu et al., 2019) that exploits egocentric videos and BEV pictures. However, the results unveil that both models still largely fall behind human performances by a large margin (47.2% of the best model vs. 90.06% of amateur human testers). To understand the failure modes, we conduct experiments on settings that could alleviate the challenges brought by situation understanding. The improvement of these models confirms that the current models are indeed struggling with situation understanding, which is pivotal for embodied scene understanding. Finally, we explore whether powerful Large Language Models (LLMs) like GPT-3 (Brown et al., 2020) and Unified QA (Khashabi et al., 2020) could tackle our tasks by converting the multi-modal SQA3D problems into single-modal surrogates using scene captioning. However, our results read that these models can still be bottlenecked by the lack of spatial understanding and accurate captions.

Our contributions can be summarized as follow:

- We introduce SQA3D, a new benchmark for embodied scene understanding, aiming at reconciling the challenging capabilities of situation understanding and situated reasoning and facilitating the development of intelligent embodied agents.

- We meticulously curate the SQA3D to include diverse situations and interesting questions. These questions probe a wide spectrum of knowledge and reasoning abilities of embodied agents, ranging from spatial relation comprehension to navigation, common sense reasoning, and multi-hop reasoning.

- We perform extensive analysis on the state-of-the-art multi-modal reasoning models. However, experimental results indicate that these avenues are still struggling on SQA3D. Our hypothesis suggests the crucial role of proper 3D representations and the demand for better situation understanding in embodied scene understanding.

## 2 THE SQA3D DATASET

A problem instance in SQA3D can be formulated as a triplet $\langle \mathcal{S}, s, q \rangle$, where $\mathcal{S}$ denotes the scene context, *e.g.*, 3D scan, egocentric video, bird-eye view (BEV) picture, *etc.*; $s = \langle s^{\text{txt}}, s^{\text{pos}}, s^{\text{rot}} \rangle$ denotes a situation, where the textual situation description $s^{\text{txt}}$ (*e.g.*, "*Sitting at the edge of the bed and facing the couch*" in Figure 1) depicts the position $s^{\text{pos}}$ and orientation $s^{\text{rot}}$ of an agent in the scene; $q$ denotes a question. The task is to retrieve the correct answer from the answer set $a = \{a_1, \ldots, a_N\}$, while optionally predicting the ground truth location $\langle s^{\text{pos}}, s^{\text{rot}} \rangle$ from the text. The additional prediction of location could help alleviate the challenges brought by situation understanding. The following subsections will detail how to collect and curate the data and then build the benchmark.

### 2.1 DATA FORMATION

The 3D indoor scenes are selected from the ScanNet (Dai et al., 2017) dataset. We notice that some scenes could be too crowded/sparse, or overall tiny, making situations and questions collection infeasible. Therefore, we first manually categorize these scenes based on the richness of objects/layouts and the space volume. We end up retaining 650 scenes after dropping those that failed to meet the requirement. We then develop an interactive web-based user interface (UI) to collect the data. Details of UI design can be found in *appendix*. All the participants are recruited on AMT.

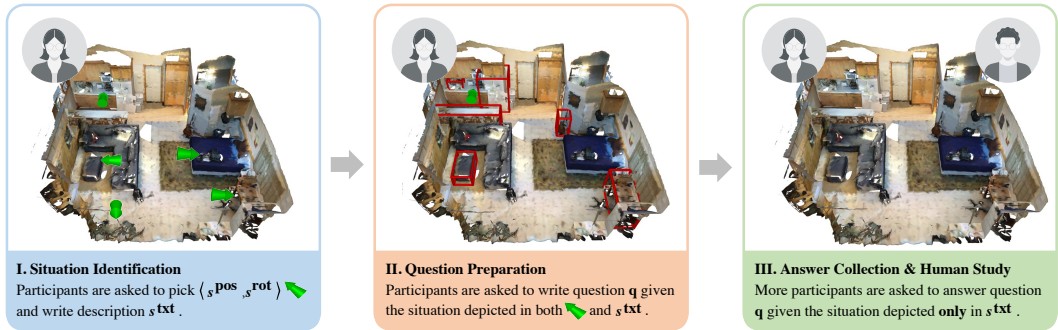

Figure 3: **Data collection pipeline of SQA3D.** Since our dataset comprises multiple types of annotations (situations and their descriptions, questions, answers, *etc.*), we found it more manageable to break down a single annotation task into three sub-tasks: i) Situation Identification; ii) Question Preparation; iii) Answer Collection & Human Study, where the participants recruited on AMT only need to focus on a relatively simple sub-task at a time.

Compared to counterparts, the annotation load of a single SQA3D problem instance could be significantly heavier as participants need to explore the scene, pick a situation, make descriptions, and ask a few questions. All these steps also require dense interaction with the 3D scene. To ensure good quality, we introduce a **multi-stage collection** pipeline, which breaks down the load into more manageable sub-tasks. Figure 3 delineates this process:

**I. Situation Identification.** We ask the workers to pick 5 situations by changing the location $\langle s^{\text{pos}}, s^{\text{rot}} \rangle$ of a virtual avatar in a ScanNet scene $\mathcal{S}$. The workers are then instructed to write descriptions $s^{\text{txt}}$ that can **uniquely** depict these situations in the scene. We also use examples and bonuses to encourage **more natural sentences** and the **use of human activities** (*e.g.*, "*I'm waiting for my lunch to be heated in front of the microwave*"). All the collected situations are later manually curated to ensure diversity and the least ambiguity. If necessary, we would augment the data with more situations to cover different areas of the scene.

**II. Question Preparation.** We collect a set of questions w.r.t. each pair of the 3D scene $\mathcal{S}$, and the situation description $s^{\text{txt}}$ (the virtual avatar is also rendered at $\langle s^{\text{pos}}, s^{\text{rot}} \rangle$). To help prepare questions that require **substantial situated reasoning**, we tutor the workers before granting them access to our tasks. They are instructed to follow the rules and learn from good examples. We also remove & penalize the responses that do not depend on the current situation, *e.g.* "*How many chairs are there in the room?*".

**III. Answer Collection & Human Study.** In addition to the answers collected alongside the questions, we send out the questions to more workers and record their responses. These workers are provided with the same interface as in stage **II** except showing in the scene to ensure consistency between question and answer collection. There is also **mandatory scene familiarization** in all three steps before the main job starts and we find it extremely helpful especially for more crowded scenes. More details can be found in *appendix*.

## 2.2 CURATION, DATA STATISTICS, AND METRICS

**Curation.** Our multi-stage collection ends up with around 21k descriptions of 6.8k unique situations and 35k questions. Although the aforementioned prompt did yield many high-quality annotations, some of them are still subject to curation. We first apply a basic grammar check to clean up the language glitches. Then we follow the practices in VQAv2 (Goyal et al., 2017) and OK-VQA (Marino et al., 2019) to further eliminate low-effort descriptions and questions. Specifically, we eliminate & rewrite template-alike descriptions (*e.g.*, repeating the same sentence patterns) and questions that are too simple or do not require looking at the scene. We also notice the similar answer bias reported in Marino et al. (2019) where some types of questions might bias toward certain answers. Therefore, we remove questions to ensure a more uniform answer distribution. A comparison of answer distribution before and after the balancing can be found in *appendix*. As a result, our final dataset comprises 20.4k descriptions and 33.4k diverse and challenging questions. Figure 2 demonstrates some example questions in SQA3D.

**Statistics.** Compared to most counterparts with template-based text generation, SQA3D is crowd-sourced on AMT and therefore enjoys more naturalness and better diversity. To the best of our knowledge, SQA3D is the **largest** dataset of grounded 3D scene understanding with the human-annotated question-answering pairs (a comparison to the counterparts can be found in Table 1).

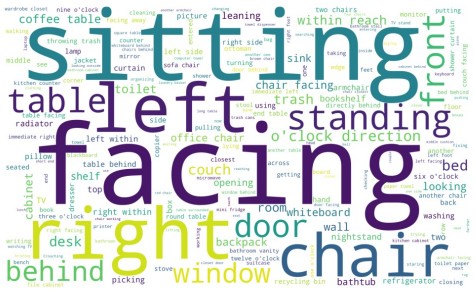

Figure 4: Word cloud of $s^{\text{txt}}$ in SQA3D.

| Statistic | Value |
|---|---|
| Total $s^{\text{txt}}$ (train/val/test) | 16,229/1,997/2,143 |
| Total $q$ (train/val/test) | 26,623/3,261/3,519 |
| Unique $q$ (train/val/test) | 20,183/2,872/3,036 |
| Total scenes (train/val/test) | 518/65/67 |
| Total objects (train/val/test) | 11,723/1,550/1,652 |
| Average $s^{\text{txt}}$ length | 17.49 |
| Average $q$ length | 10.49 |

Table 2: SQA3D dataset statistics.

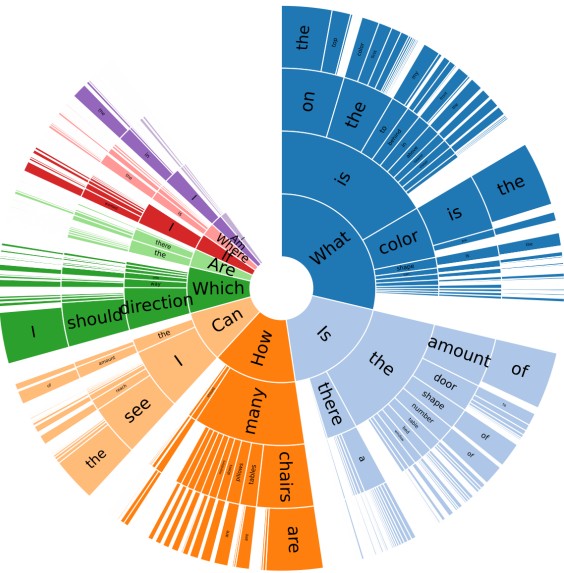

Figure 5: Question distribution in SQA3D

Table 2, Figure 4, and Figure 5 illustrate the basic statistics of our dataset, including the word cloud of situation descriptions and question distribution based on their prefixes. It can be seen that descriptions overall meet our expectations as human activities like "sitting" and "facing" are among the most common words. Our questions are also more diverse and balanced than our counterparts, where those starting with "What" make up more than half of the questions and result in biased questions (Azuma et al., 2022). More statistics like distributions over answers and length of the text can be found in *appendix*.

**Dataset splits and evaluation metric.** We follow the practice of ScanNet and split SQA3D into *train*, *val*, and *test* sets. Since we cannot access the semantic annotations in ScanNet *test* set, we instead divide the ScanNet validation scenes into two subsets and use them as our *val* and *test* sets, respectively. The statistics of these splits can be found in Table 2. Following the protocol in VQAv2 (Goyal et al., 2017), we provide a set of 706 "top-K" answer candidates by excluding answers that only appear very few times. Subsequently, we adopt the "exact match" as our evaluation metric, *i.e.*, the accuracy of answer classification in the *test* set. No further metric is included as we find it sufficient enough to measure the differences among baseline models with "exact match".

## 3 MODELS FOR SQA3D

Generally speaking, SQA3D can be characterized as a multi-modal reasoning problem. Inspired by the recent advances in transformer-based (Vaswani et al., 2017) vision-language models (Lu et al., 2019; Li et al., 2020; Alayrac et al., 2022), we investigate how could these methods approach our task. Specifically, we study a recent transformer-based question-answering system: ScanQA (Azuma et al., 2022), which maps 3D scans and questions into answers. We make a few adaptations to ensure its compatibility with the protocol in SQA3D. To further improve this model, we consider including some auxiliary tasks during training (Ma et al., 2022). For other types of 3D scene context, *e.g.* egocentric video clips and BEV pictures, we employ the corresponding state-of-the-art models. Finally, we explore the potential of recently-introduced LLMs like GPT-3 (Brown et al., 2020) and Unified QA (Khashabi et al., 2020) on solving SQA3D in a zero-shot fashion. An overview of these models can be found in Figure 6.

**3D model.** We use the term *3D model* to refer a modified version of the ScanQA model (Azuma et al., 2022), depicted in the blue box of Figure 6. It includes a VoteNet (Qi et al., 2019)-based 3D perception module that extracts object-centric features, LSTM-based language encoders for processing both questions $q$ and situation description $s^{\text{txt}}$, and some cross-attention transformer blocks (Vaswani et al., 2017). The object-centric feature tokens attend to the language tokens of $s^{\text{txt}}$ and $q$ successively. Finally, these features will be fused and mapped to predict the answer. Optionally, we can add one head to predict the location $\langle s^{\text{pos}}, s^{\text{rot}} \rangle$ of the agent. Since the VoteNet module is trained from scratch, we also employ an object detection objective (not shown in the figure).

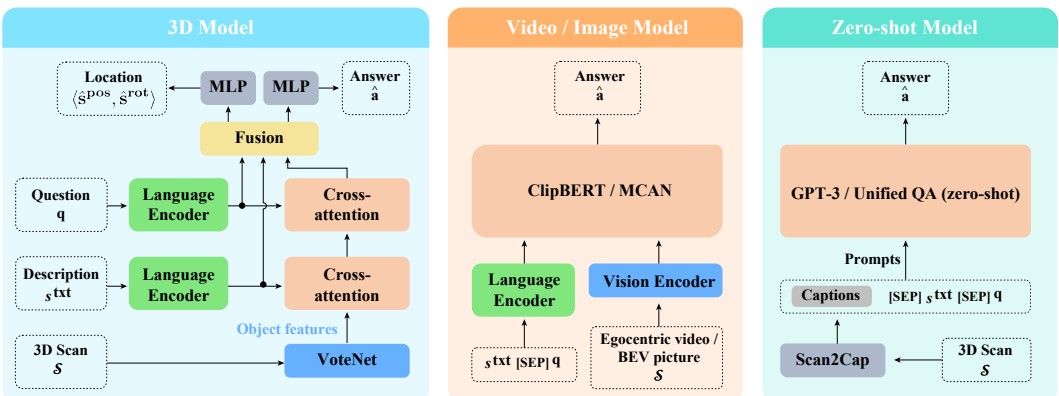

Figure 6: **Potential models for SQA3D.** We split the considered models into three groups: 3D model, video / image model, and zero-shot model. The 3D model is modified from the ScanQA model (Azuma et al., 2022) and maps 3D scan input to the answer. While the video / image models are effectively borrowed from canonical video QA and VQA tasks but we augment them with the additional situation input. The zero-shot model explores the potential of large pre-trained LLMs on our tasks. But they have to work with an additional 3D caption model that converts the 3D scene into text.

**Auxiliary task.** As we mentioned before, situation understanding plays a crucial role in accomplishing SQA3D tasks. To encourage a better understanding of the specified situation, we introduce two auxiliary tasks: the model is required to make predictions about the $s^{\text{pos}}$ and $s^{\text{rot}}$ of the situation. We use mean-square-error (MSE) loss for these tasks. The overall loss for our problem therefore becomes $\mathcal{L} = \mathcal{L}_{\text{ans}} + \alpha\mathcal{L}_{\text{pos}} + \beta\mathcal{L}_{\text{rot}}$, where $\mathcal{L}_{\text{ans}}$, $\mathcal{L}_{\text{pos}}$, and $\mathcal{L}_{\text{rot}}$ depicts the losses of the main and auxiliary tasks, $\alpha$ and $\beta$ are balancing weights.

**Video and Image-based model.** The orange box in the middle of Figure 6 demonstrates the models for video and image-based input. SQA3D largely resembles a video question answering or visual question answering problem when choosing to represent the 3D scene context $\mathcal{S}$ as egocentric video clips or BEV pictures. However, SQA3D also requires the model to take both question $q$ and the newly added situation description $s^{\text{txt}}$ as input. We, therefore, follow the practice in the task of context-based QA (Rajpurkar et al., 2018) and prepend $s^{\text{txt}}$ to the question as a *context*. For the model, we use the state-of-the-art video QA system ClipBERT (Lei et al., 2021) and VQA system MCAN (Yu et al., 2019). We adopt most of their default hyper-parameters and the details can be found in *appendix*.

**Zero-shot model.** We explore to which extent the powerful LLMs like GPT-3 (Brown et al., 2020) and Unified QA (Khashabi et al., 2020) could tackle our tasks. Following prior practices that apply GPT-3 to VQA (Changpinyo et al., 2022; Gao et al., 2022), we propose to convert the 3D scene into text using an emerging technique called 3D captioning (Chen et al., 2021). We provide the caption, $s^{\text{txt}}$, and $q$ as part of the prompt and ask these models to complete the answer. For GPT-3, we further found providing few-shot examples in the prompt helpful with much better results. Minor post-processing is also needed to ensure answer quality. We provide more details on prompt engineering in the *appendix*.

## 4 EXPERIMENTS

### 4.1 SETUP

We benchmark the models introduced in Section 3 to evaluate their performances on SQA3D. As mentioned before, we examine three types of scene context $\mathcal{S}$: 3D scan (point cloud), egocentric video, and BEV picture. Both the 3D scan and egocentric video for each scene are provided by ScanNet (Dai et al., 2017). However, we down-sample the video to allow more efficient computation per the requirement of the ClipBERT model (Lei et al., 2021). The BEV pictures are rendered by placing a top-down camera on top of the scan of each 3D scene. We also conduct additional experiments that investigate factors that could contribute to the results, *e.g.*, situation and auxiliary tasks. In our early experiments, we found that the 3D model overall performs better than the video or image-based models. Therefore we only conduct these additional experiments with the variants of our 3D model due to the limit of computational resources. We use the official implementation of ScanQA, ClipBERT, and MCAN and include our modifications for SQA3D. For the zero-shot models, we extract 3D scene captions from two sources: ScanRefer (Chen et al., 2020) and ReferIt3D (Achlioptas et al., 2020). Considering the limit on the length of the input prompt,

| | $\mathcal{S}$ | Format | test set | | | | | | Avg. |
|---|---|---|---|---|---|---|---|---|---|
| | | | What | Is | How | Can | Which | Others | |
| Blind test | - | SQ→A | 26.75 | 63.34 | 43.44 | **69.53** | 37.89 | 43.41 | 43.65 |
| ScanQA (w/o $s^{\text{txt}}$) | 3D scan | VQ→A | 28.58 | 65.03 | **47.31** | 66.27 | 43.87 | 42.88 | 45.27 |
| ScanQA | 3D scan | VSQ→A | 31.64 | 63.80 | 46.02 | 69.53 | 43.87 | 45.34 | 46.58 |
| ScanQA + aux. task | 3D scan | VSQ→AL | 33.48 | **66.10** | 42.37 | 69.53 | 43.02 | **46.40** | **47.20** |
| MCAN | BEV | VSQ→A | 28.86 | 59.66 | 44.09 | 68.34 | 40.74 | 40.46 | 43.42 |
| ClipBERT | Ego. video | VSQ→A | 30.24 | 60.12 | 38.71 | 63.31 | 42.45 | 42.71 | 43.31 |
| Unified QA$_{\text{Large}}$ | ScanRefer | VSQ→A | 33.01 | 50.43 | 31.91 | 56.51 | **45.17** | 41.11 | 41.00 |
| Unified QA$_{\text{Large}}$ | ReferIt3D | VSQ→A | 27.58 | 47.99 | 34.05 | 59.47 | 40.91 | 39.77 | 38.71 |
| GPT-3 | ScanRefer | VSQ→A | **39.67** | 45.99 | 40.47 | 45.56 | 36.08 | 38.42 | 41.00 |
| GPT-3 | ReferIt3D | VSQ→A | 28.90 | 46.42 | 28.05 | 40.24 | 30.11 | 36.07 | 34.57 |
| Human (amateur) | 3D scan | VSQ→A | 88.53 | 93.84 | 88.44 | 95.27 | 87.22 | 88.57 | 90.06 |

Table 3: **Quantitative results on the SQA3D benchmark**. Results are presented in accuracy (%) on different types of questions. In the "Format" column: V = 3D visual input $\mathcal{S}$; S = situation description $s^{\text{txt}}$; Q = question $q$; A = answer $a$; L = location $\langle s^{\text{pos}}, s^{\text{rot}} \rangle$. In ScanQA, *aux. task* indicates the use of both $\mathcal{L}_{\text{pos}}$ and $\mathcal{L}_{\text{rot}}$ as additional losses. We use the *Large* variant as Unified QA (Khashabi et al., 2020) as it works better.

these 3D captions are also down-sampled. The Unified QA model weights are obtained from its Huggingface official repo. All the models are tuned using the validation set and we only report results on the test set. More details on model implementation can be found in *appendix*.

## 4.2 QUANTITATIVE RESULTS

We provide the quantitative results of the considered models (detailed in Section 3) on our SQA3D benchmark in Table 3. The findings are summarized below:

**Question types.** In Table 3, we demonstrate accuracy on six types of questions based on their prefixes. Most models tend to perform better on the "Is" and "Can" questions while delivering worse results on "What" questions, likely due to a smaller number of answer candidates – most questions with binary answers start with "Is" and "Can", offering a better chance for the random guess. Moreover, we observe the hugest gap between the blind test (model w/o 3D scene context input) and our best model on the "What" and "Which" categories, suggesting the need for more visual information for these two types of questions. This also partially echoes the finding reported in Lei et al. (2018).

**Situation understanding and reasoning.** At the heart of SQA3D benchmark is the requirement of situation understanding and reasoning. As we mentioned in Section 2.1, the model will be more vulnerable to wrong answer predictions if ignoring the situation that the question depends on (*e.g.* "*What is in front of me*" could have completely different answers under different situations). In Table 3, removing situation description $s^{\text{txt}}$ from the input leads to worse results, while adding the auxiliary situation prediction tasks boosts the overall performance, especially on the challenging "What" questions. The only exception is "How" questions, where a majority of them are about counting. We hypothesize that most objects in each ScanNet scene only have a relatively small number of instances, and the number could also correlate to the object category. Therefore, guessing/memorization based on the question only could offer better results than models with the situation as input if the situation understanding & reasoning are still not perfect yet. Additionally, we also provide an inspection of the relation between situation understanding and QA using attention visualization in Section 4.3.

**Representations of 3D scenes.** Indeed, SQA3D does not limit the input to be 3D scan only, as we also offer options of egocentric videos and BEV pictures. Compared to models with the 3D scan as input, the tested models with other 3D representations (*i.e.*, MCAN and ClipBERT) deliver much worse results, implying that the 3D scan so far could still be a better representation for the 3D scene when the reasoning models are probed with questions that require a holistic understanding of the scene. On the other hand, MCAN and ClipBERT are general-purpose QA systems, while ScanQA is designed for 3D-language reasoning tasks. The generalist-specialty trade-off could also partially account for the gap. Finally, the poor results of BEV and egocentric videos based models compared to the blind test could also be due to the additional "vision-bias" when the visual input is provided (Antol et al., 2015). Note that the vision-bias can be mitigated with better visual representations (Wen et al., 2021), implying that ScanQA, which seems to suffer less from the vision-bias

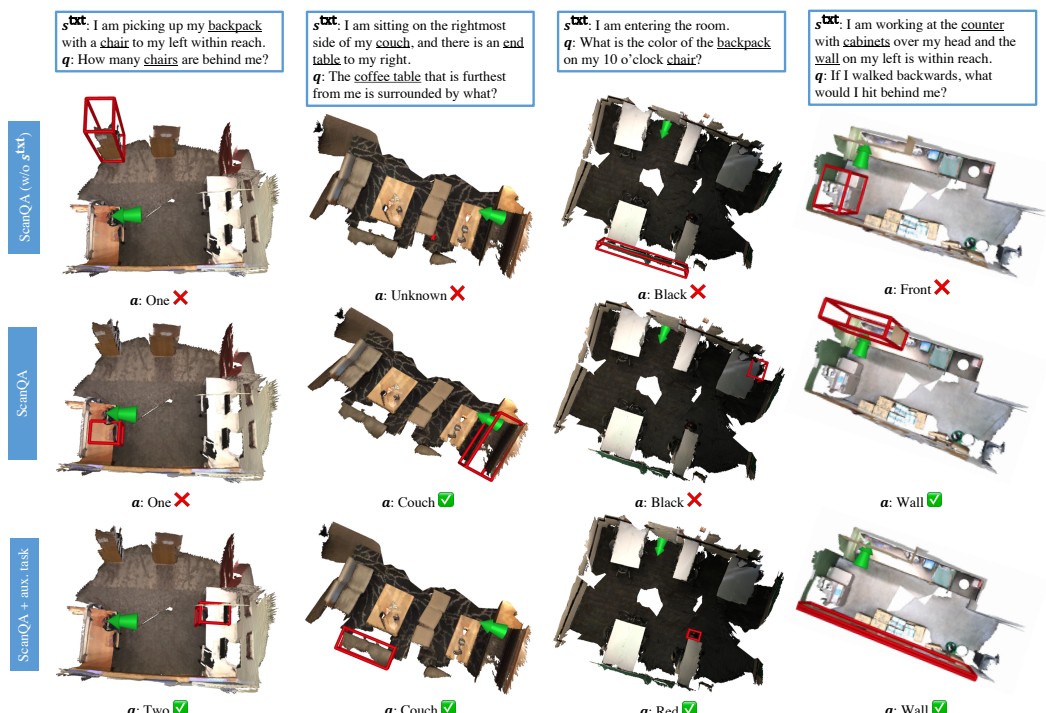

Figure 7: **Qualitative results.** We show the predicted answer and bbox with highest attention for the variants of ScanQA (Azuma et al., 2022) models. We anticipate the bbox to indicate the object that situation description $s^{txt}$ or question $q$ refers to. We observe that better situation understanding (via comprehension on $s^{txt}$ or auxiliary tasks) could result in more reasonable attention over objects, which positively correlates to more robust answer prediction.

than the counterparts using BEV and egocentric videos, is fueled by better visual representations in terms of combating the dataset bias.

**Zero-shot vs. training from scratch.** The success of pre-trained LLMs like GPT-3 on myriads of challenging reasoning tasks (Wei et al., 2022b;a) suggests that these models could possibly also understand embodied 3D scenes with language-only input (Landau & Jackendoff, 1993). However, SQA3D imposes a grand challenge to these models. The powerful Unified QA (*Large* variant) and GPT-3 both fail to deliver reasonable results on our tasks. Further, we hypothesize the bottleneck could also be on the 3D captions, as the results verify the consistent impact on model performances brought by a different source of captions (ScanRefer→ReferIt3D). However, we still believe these models have great potential. For example, one zero-shot model (GPT-3 + ScanRefer) do pretty well on the challenging "What" questions (39.67%), even better than the best ScanQA variant.

**Human vs. machine.** Finally, all the machine learning models largely fall behind amateur human participants (47.2% of ScanQA + aux. task vs. 90.06%). Notably, we only offer a limited number of examples for the testers before sending them the SQA3D problems. Our participants promptly master how to interact with the 3D scene, understand the situation from the textual description, and answer the challenging questions. The human performance also shows no significant bias for different question types.

### 4.3 QUALITATIVE RESULTS

Finally, we offer some qualitative results of the variants of our 3D model in Figure 7. We primarily focus on visualizing both the answer predictions and the transformer attention over the object-centric feature tokens (bounding boxes) generated by the VoteNet (Qi et al., 2019) backbone. We highlight the most-attended bounding box among all the predictions by the transformer-based model, in the hope of a better understanding of how these models perceive the 3D scene to comprehend the situations and answer the questions. In Figure 7, the correct predictions are always associated with attention over relevant objects in the situation description $s^{txt}$ and questions. Moreover, in case there are multiple instances of the same object category, it is also crucial to identify the correct instance. For example, only ScanQA + aux. task makes the correct prediction for the first question and also attends to the right chair behind 🟢, while ScanQA focuses on a wrong instance. These results con-

firm our findings in Section 4.2 about the critical role of situation understanding. We also provide some failure modes in *appendix*.

## 5 RELATED WORK

**Embodied AI.** The study of embodied AI (Brooks, 1990) emerges from the hypothesis of "*ongoing physical interaction with the environment as the primary source of constraint on the design of intelligent systems*". To this end, researchers have proposed a myriad of AI tasks to investigate whether intelligence will emerge by acting in virtual or photo-realistic environments. Notable tasks including robotic navigation (Das et al., 2018; Anderson et al., 2018; Savva et al., 2019; Chen et al., 2019; Wijmans et al., 2019b; Qi et al., 2020; Deitke et al., 2022) and vision-based manipulation (Kolve et al., 2017; Puig et al., 2018; Xie et al., 2019; Shridhar et al., 2020a;b; 2022). These tasks are made more challenging as instructions or natural-dialogues are further employed as conditions. Sophisticated models have also been developed to tackle these challenges. Earlier endeavors usually comprise multi-modal fusion (Tenenbaum & Freeman, 1996; Perez et al., 2018) and are trained from scratch (Wang et al., 2018; Fried et al., 2018; Wang et al., 2019), while recent efforts would employ pre-trained models (Pashevich et al., 2021; Hong et al., 2021a; Suglia et al., 2021). However, the agents still suffer from poor generalization to novel and more complex testing tasks (Shridhar et al., 2020a) compared to results on training tasks. More detailed inspection has still yet to be conducted and it also motivates our SQA3D dataset, which investigates one crucial capability that the current embodied agents might need to improve: **embodied scene understanding**.

**Grounded 3D understanding.** Visual grounding has been viewed as a key to connecting human knowledge, which is presumably encoded in our language, to the visual world, so as enable the intelligent agent to better understand and act in the real environment. It is natural to extend this ability to 3D data as it offers more immersive representations of the world. Earlier work has examined word-level grounding with detection and segmentation tasks on 3D data (Gupta et al., 2013; Song & Xiao, 2014; Dai et al., 2017; Chang et al., 2017). Recent research starts to cover sentence-level grounding with complex semantics (Chen et al., 2020; Achlioptas et al., 2020; Chen et al., 2021). More recently, new benchmarks introduce complex visual reasoning to 3D data (Azuma et al., 2022; Ye et al., 2021; Yan et al., 2021). However, these tasks mostly assume a passive, third-person's perspective, while our SQA3D requires problem-solving with an egocentric viewpoint. This introduces both challenges and chances for tasks that need a first-person's view, *e.g.* embodied AI.

**Multi-modal question answering.** Building generalist question answering (QA) systems has long been a goal for AI. Along with the progress in multi-modal machine learning, VQA (Antol et al., 2015; Zhu et al., 2016) pioneers the efforts of facilitating the development of more human-like, multi-modal QA systems. It has been extended with more types of knowledge, *e.g.* common sense (Zellers et al., 2019) and factual knowledge (Marino et al., 2019). Recent research has also introduced QA tasks on video (Lei et al., 2018; Jia et al., 2020; 2022; Grunde-McLaughlin et al., 2021; Wu et al., 2021; Datta et al., 2022), and 3D data (Ye et al., 2021; Azuma et al., 2022; Yan et al., 2021). We propose the SQA3D benchmark also in hope of facilitating multi-modal QA systems with the ability of embodied scene understanding. Notably, models for SQA3D could choose their input from a 3D scan, egocentric video, or BEV picture, which makes our dataset compatible with a wide spectrum of existing QA systems.

## 6 CONCLUSION

We've introduced SQA3D, a benchmark that investigates the capability of embodied scene understanding by combining the best of situation understanding and situated reasoning. We carefully curate our dataset to include diverse situations and interesting questions while preserving the relatively large scale (20.4k situation descriptions and  33.4k questions). Our questions probe a wide spectrum of knowledge and reasoning abilities of embodied agents, notably navigation, common sense, and multi-hop reasoning. We examine many state-of-the-art multi-modal reasoning systems but the gap between the best ML model and human performances so far is still significant. Our findings suggest the crucial role of proper 3D representations and better situation understanding. With SQA3D, we hope of fostering research efforts in developing better embodied scene understanding methods and ultimately facilitate the emergence of more intelligent embodied agents.

ACKNOWLEDGEMENT

The authors would like to thank Dave Zhenyu Chen for his insightful ScanRefer project and help on data collection, Wenjuan Han for discussions on data collection and model design. This project is supported by National Key R&D Program of China (2021ZD0150200).

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

# A DATA COLLECTION

## A.1 DATA COLLECTION WEB UI

We present the Web UI of our data collection in Figure 8 (Stage I), Figure 9 (Stage II) and Figure 11 (Stage III) respectively. We developed our UI based on Chen et al. (2020). These UIs share some common components: a 3D scene viewer, where the user can drag, rotate, and zoom in/out the scene; clickable objects/tags, where users might click on either the object mesh directly or the tag on the sidebar to highlight it in the scene; and an instruction set that guide the user through the task. Users may also switch between a full scene or object mesh only to focus on the tasks. The users are also required to submit multiple responses with the same scene.

Notably, we create detailed tutorials for each stage (not shown in the UI) with examples and animated demonstrations. We found tutorials and instruction sets with clear criteria on rejection and bonus(*e.g.* Figure 10) helpful with high-quality data. Finally, all the testers need to pass a test before the qualification for our task is granted.

## A.2 DATA POST-PROCESSING

There are two major data post-processing steps in SQA3D: **cleaning** and **balancing**. For cleaning, we primarily focus on grammatical correction. We adopt both rule-based cleaning and an ML-based tool called GECToR (Omelianchuk et al., 2020) in our grammatical correction pipeline. We adjust the correction threshold based on human judgment over the corrected data samples.

In the balancing step, our goal is to reduce the question-answer bias in the dataset. Therefore we follow the practice in Antol et al. (2015); Marino et al. (2019) and re-sample the questions based on their prefixes and answer type, in hope of a more balanced answer distribution. We provide answer distribution before and after balancing in Section B.1.

## A.3 MORE MTURK DETAILS

We provide the detailed MTurk job settings below:

**Region.** We enable access to our tasks in the following countries/regions:

> US, DE, GB, AU, CA, SG, NZ, NO, SE, FI, DK, IE

**Approval rate & Number of approved jobs.** The testers are required to have at least a 95% approval rate and have completed more than 1000 tasks. However, we relax this requirement to a 90% approval rate for Stage III as it is simpler than the other annotation tasks.

**Reward.** The participants will be rewarded $0.5 for each task in Stage I and II, and $0.2 for the QA tasks in Stage III, with a possibility of a bonus depending on the overall quality. We actively monitor the response quality and send bonuses/rejections daily. Note that we collected 5 responses for each task in all three stages.

**Task lifetime.** We set the lifetime as 10 days for tasks in Stage I and 20 days for those in Stage II and III. However, we found most of the tasks can be completed in less than 7 days.

# B DATASET DETAILS

## B.1 MORE STATISTICS

We provide the histogram of the answer distribution before & after balancing in Figure 12 and Figure 13, respectively. It can be seen that we manage to ensure there is no single answer that dominates any type of question (categorized by their prefixes). However, we do acknowledge that prefix-based balancing might still not be sufficient since models could also learn to use the n-grams pattern. A more effective avenue is collecting more questions with less-frequent answers, which we leave as future work.

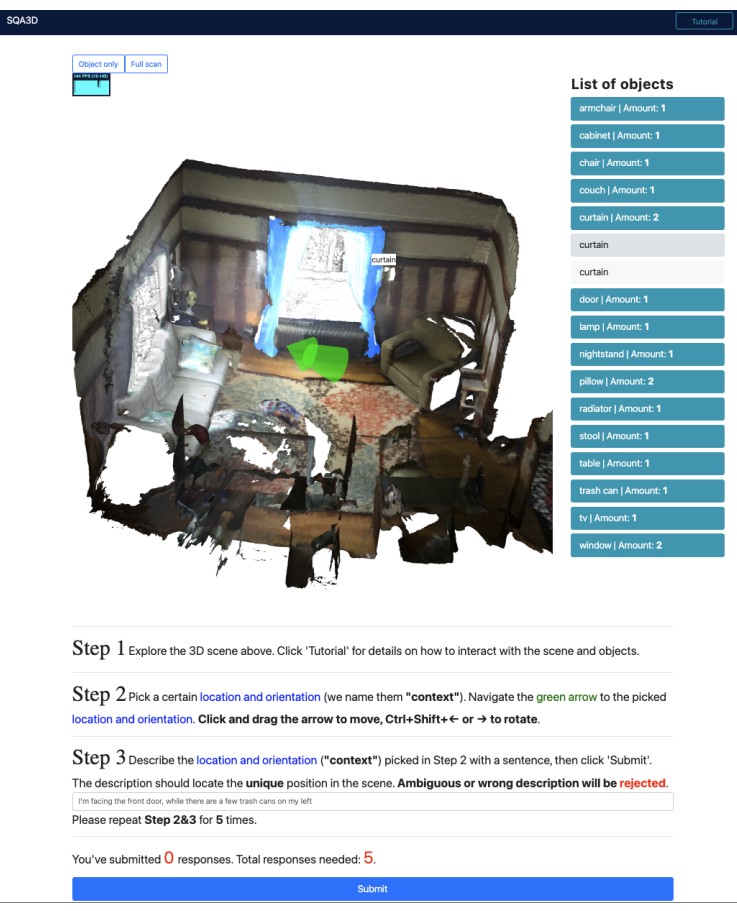

Figure 8: Dataset collection Web UI for Stage I.

In Figure 14a and Figure 14b, we show the histogram of the length of situation description $s^{\text{txt}}$ and question $q$. Overall most of the descriptions and questions are middle-length sentences (10-20 words).

## B.2 DETAILS ON EGOCENTRIC VIDEO AND BEV IMAGE

For egocentric videos, we uniformly downsample the frames of the original ScanNet (Dai et al., 2017) video by using the first frame of every 20 frames. Afterward, we resize all the frames to $224 \times 224$ to create the video used for training ClipBERT(Lei et al., 2021). Blender is used for rendering all BEV images. We compute the radius of the bounding sphere of the scene and put the camera at the top of the scene with a distance of 7 times the radius to the center of the bounding sphere. Images of size $1920 \times 1080$ are rendered for clarity while the input to the MCAN(Yu et al., 2019) model is the resized version of the images to $224 \times 224$.

## C MODEL DETAILS

### C.1 INPUT PIPELINE

We follow the input pipeline in ScanQA(Azuma et al., 2022) without further modification. As for MCAN, we only transform the images to fit the ImageNet-pretrained encoder. In ClipBERT, we randomly sample 8 clips with each clip consisting of 2 frames of the video to feed into the model as the scene representation. Note that each frame is resized to $1000 \times 1000$ following the practice of original ClipBERT(Lei et al., 2021).

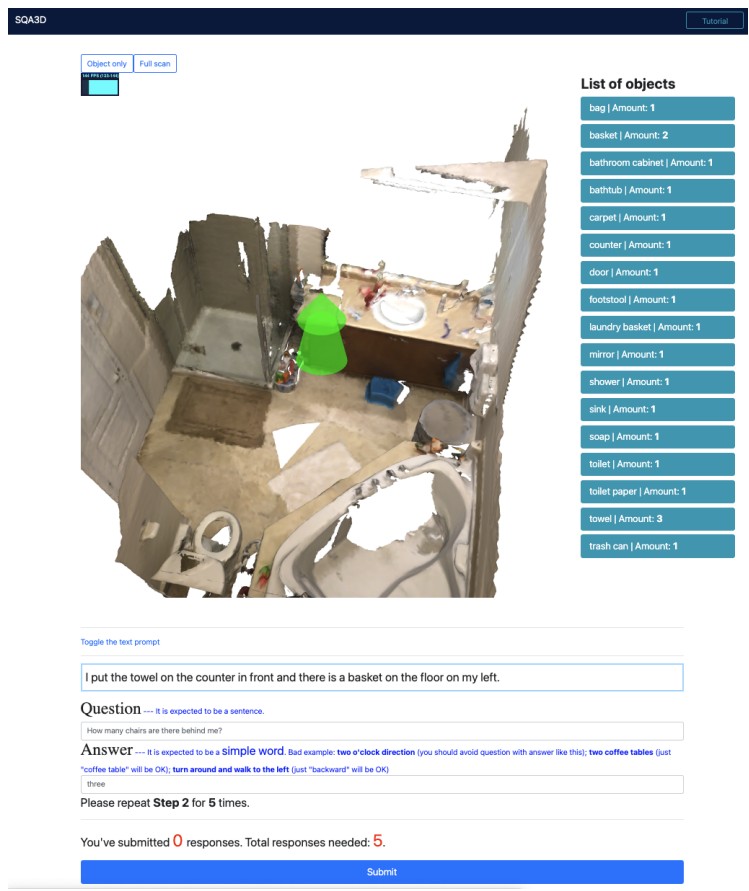

Figure 9: Dataset collection Web UI for Stage II.

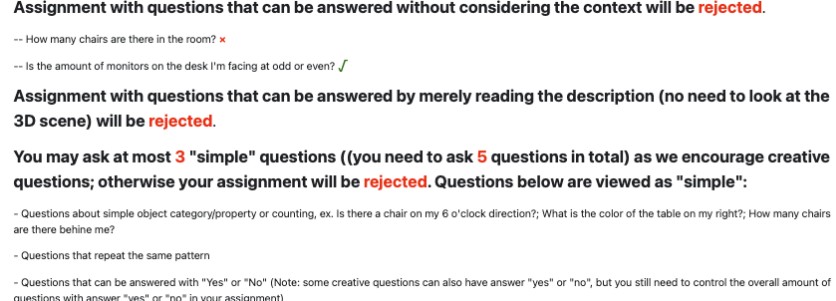

Figure 10: Additional instruction set to the AMT participants in Stage II.

## C.2    HYPER-PARAMETERS

We provide the hyper-parameters of the considered models in Table 4.

## C.3    ADDITIONAL DETAILS ON ZERO-SHOT MODELS

We uniformly sample 30 sentences from our 3D caption sources for both models. When testing with the Unified QA$_{\text{Large}}$ model, we employ a simple greedy sampling method and the following prompt:

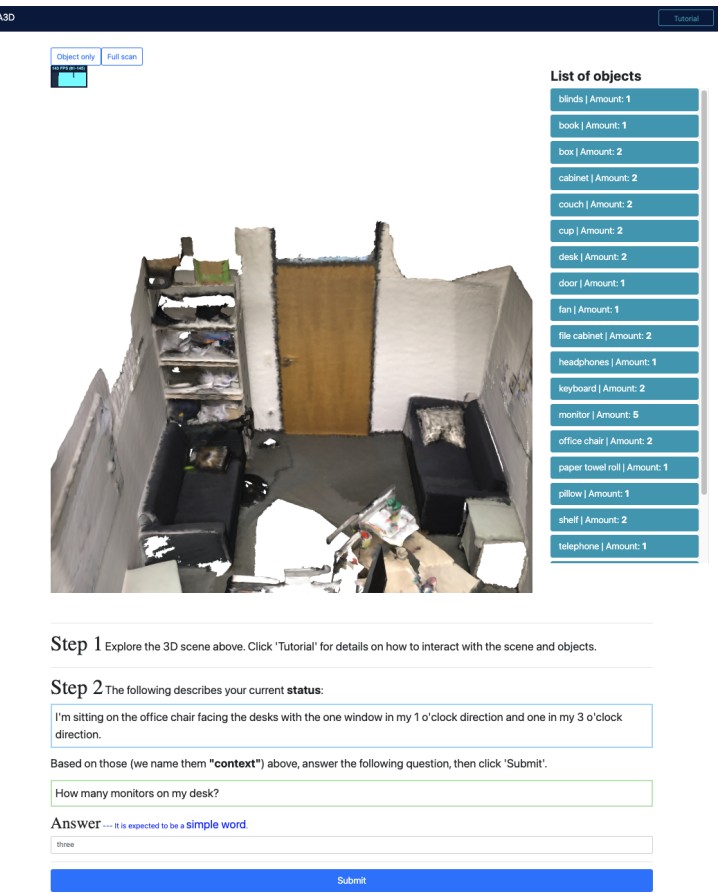

Figure 11: Dataset collection Web UI for Stage III.

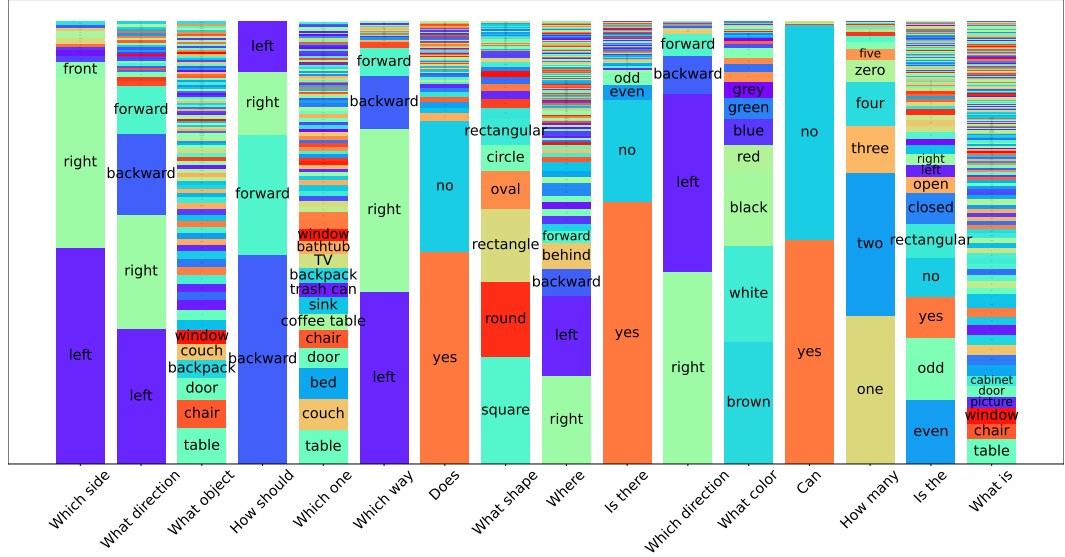

Figure 12: Answer distribution (organized by question prefixes) before balancing.

$\{s^{\text{txt}}\}$
Q: $\{q\}$
A:

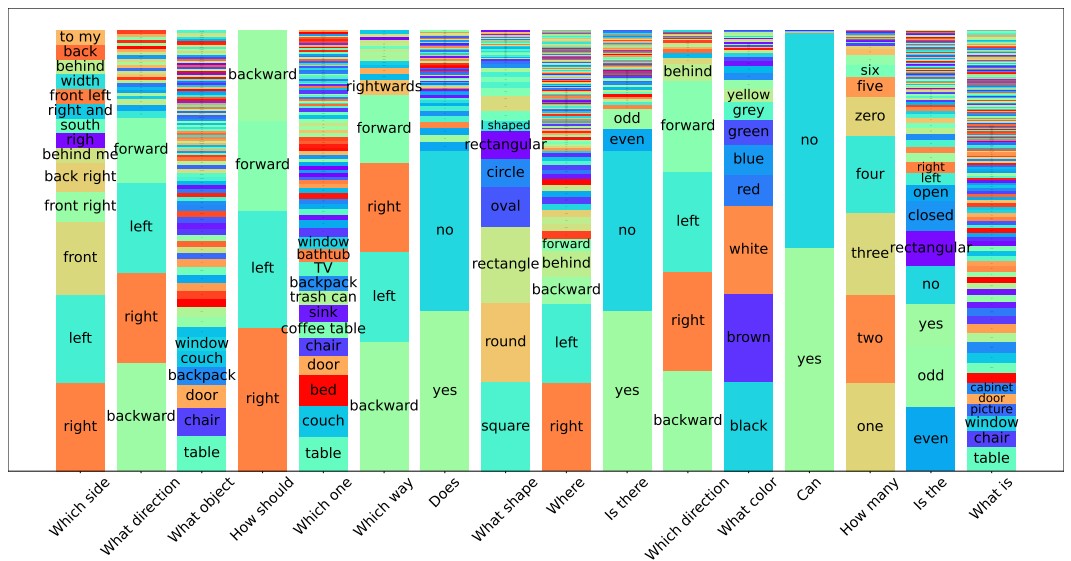

Figure 13: Answer distribution (organized by question prefixes) after balancing.

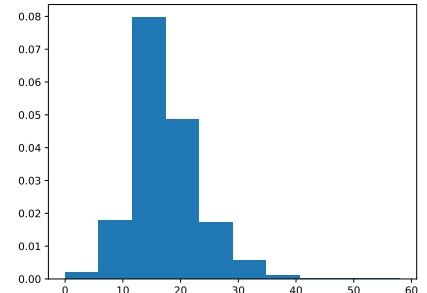

(a) Histogram of situation description $s^{\text{txt}}$ length.

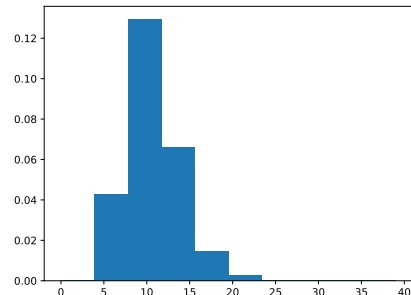

(b) Histogram of question $q$ length.

, where $\{s^{\text{txt}}\}$ and $\{q\}$ are replaced by the situation description and question. For GPT-3, we use the `text-davinci-002` variant and the following prompt:

```
Context:  There is a book on the desk.  A laptop with a green cover
is to the left of the book.
Q: I'm working by the desk.  What is on the desk beside the book?
A: laptop
Context:{s^txt}
Q: {q}
A:
```

, where we use a 1-shot example to demonstrate the format of our task. Interestingly, we found only GPT-3 would benefit from few-shot examples.

### C.4 ADDITIONAL DETAILS ON SCANQA/MCAN/CLIPBERT

**ScanQA** (Azuma et al., 2022). We slightly modify the original ScanQA code base (from https://github.com/ATR-DBI/ScanQA) to make it fit our task better. The original reference branch is discarded and the supervision signal for the language classification branch is changed to make use of it as a regression branch. More specific details can be found below.

- The original data loader only outputs the question as a whole (meaning that the situation is concatenated before the question), while our version split the two sentences.
- The original model takes language as 1 input, while we feed situation and question separately into the model.
- The original model uses 1 self-attention block and 1 cross-attention block for the fusion of language and visual features, while our version uses 2 self-attention blocks and 2 cross-attention blocks to treat situations and questions separately.
- The original model uses additive operation to fuse language & visual features, while our version uses concatenation for fusion.
- To conduct the ablation experiment of blind test, we simply discard the output feature of VoteNet and only feed the situation feature and question feature into the QA head.
- To conduct the ablation experiment of w/o $s^{\text{txt}}$, we replace situation with several $\langle unk \rangle$ tokens to make a fair comparison.
- To add an auxiliary task into training, we change the supervision of the language classification head from Cross Entropy to MSE Loss to make it a regression head.

**MCAN** (Yu et al., 2019). We use the code base from RelVIT (Ma et al., 2022) (https://github.com/NVlabs/RelViT) since its implementation of MCAN could take raw images as input while the original one cannot. The default training setting is kept except for learning rate decay. We cancel it to make a fair comparison with the other baselines. We concatenate the situation before the question to make them as a whole and use this new sentence as the question that MCAN requires.

**ClipBERT** (Lei et al., 2021). We use the official repository of ClipBERT (https://github.com/jayleicn/ClipBERT) and follow the instruction to transform our data into the format ClipBERT takes. The configuration file for MSR-VTT QA (Xu et al., 2016) is used for generality as we find all the configuration files to be almost identical. The evaluated question types are changed since our focus is different from MSR-VTT. We turn off mixed precision training as we observe instability when using it. We concatenate the situation before the question to make them as a whole and use this new sentence as the question that ClipBERT requires.

## D ADDITIONAL EMPIRICAL RESULTS

We provide additional qualitative results and failure modes in Figure 15 and Figure 16.

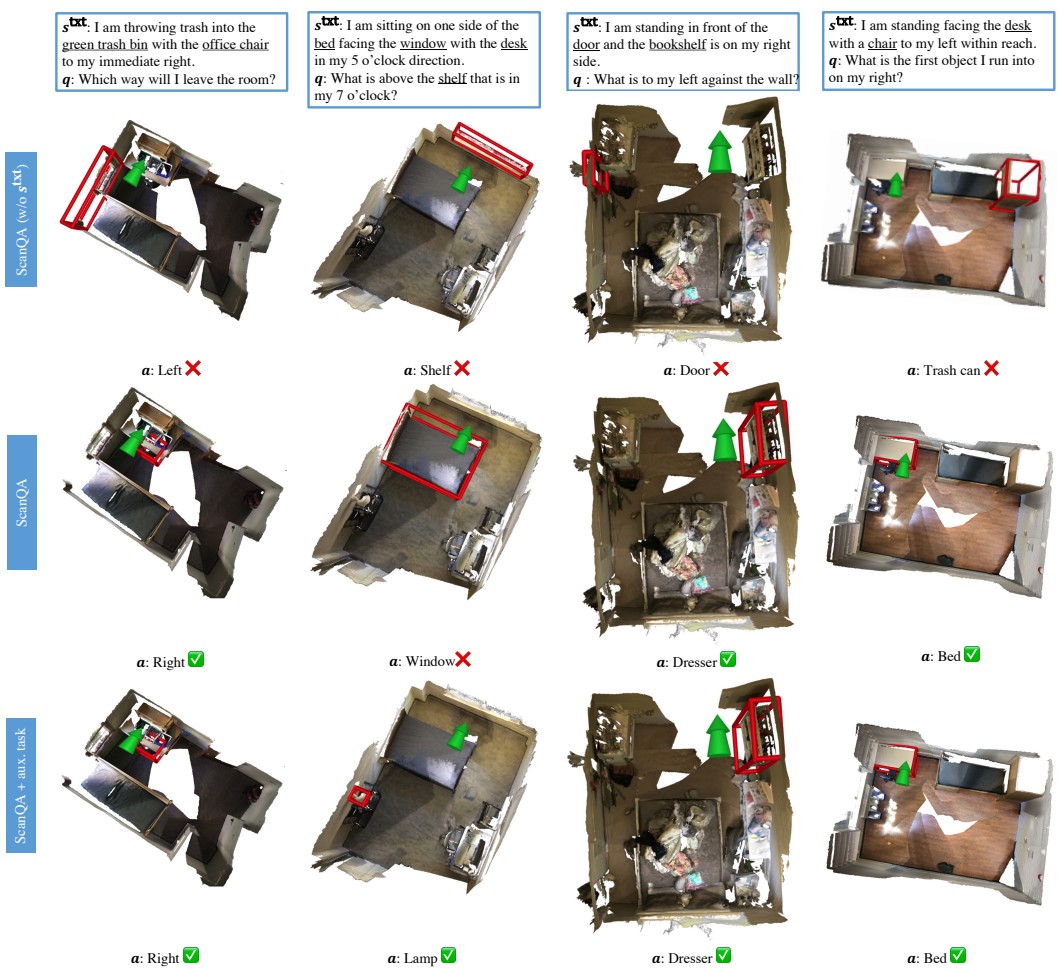

Figure 15: Additional qualitative results.

## E    LIMITATION AND POTENTIAL IMPACT

**Limitations.** One major limitation of SQA3D is the selection of 3D scenes. Since our dataset is collected on mostly indoor ScanNet scenes about household environments, it cannot cover outdoor scenes and other types of scenes, ex. warehouse. This could limit the application to autonomous driving and warehouse robots, which are likely deployed to the scene types that do not present in SQA3D. Moreover, all the scenes in ScanNet are static, *i.e.* the agent cannot interact with the object, making the exploration in SQA3D limited to hovering over the 3D scenes. However, many embodied tasks also require non-trivial interaction with articulated objects, *e.g.* drawers. Therefore, the embodied scene understanding capability examined by SQA3D can also be limited to non-interactive scenarios, *i.e.* **situation understanding** and **situated reasoning**.

**Societal impact.** SQA3D offers two sets of annotations: situations $\langle s^{\text{txt}}, s^{\text{pos}}, s^{\text{rot}} \rangle$ and QA $\langle q, a \rangle$. The situation annotations themselves could enable many exciting applications including building a real-world household assistant robot – one of its core capabilities is connecting natural language instructions/descriptions to the situations in the scene, *e.g.* locations. Moreover, non-trivial commonsense reasoning is also required in this process. Our annotations with accurate description-location pairs and the requirement of commonsense knowledge in text understanding could support these needs. The QA tasks also examine a wide spectrum of capabilities of embodied agents in household domains, making it a great benchmark for testing these household assistant robots. Finally, we will also release all the annotation interfaces and meta information, inviting everyone from either academia or industry to develop a customized version of QA datasets upon SQA3D and its infrastructure, which might help with the development of the 3D-language-related research and products.

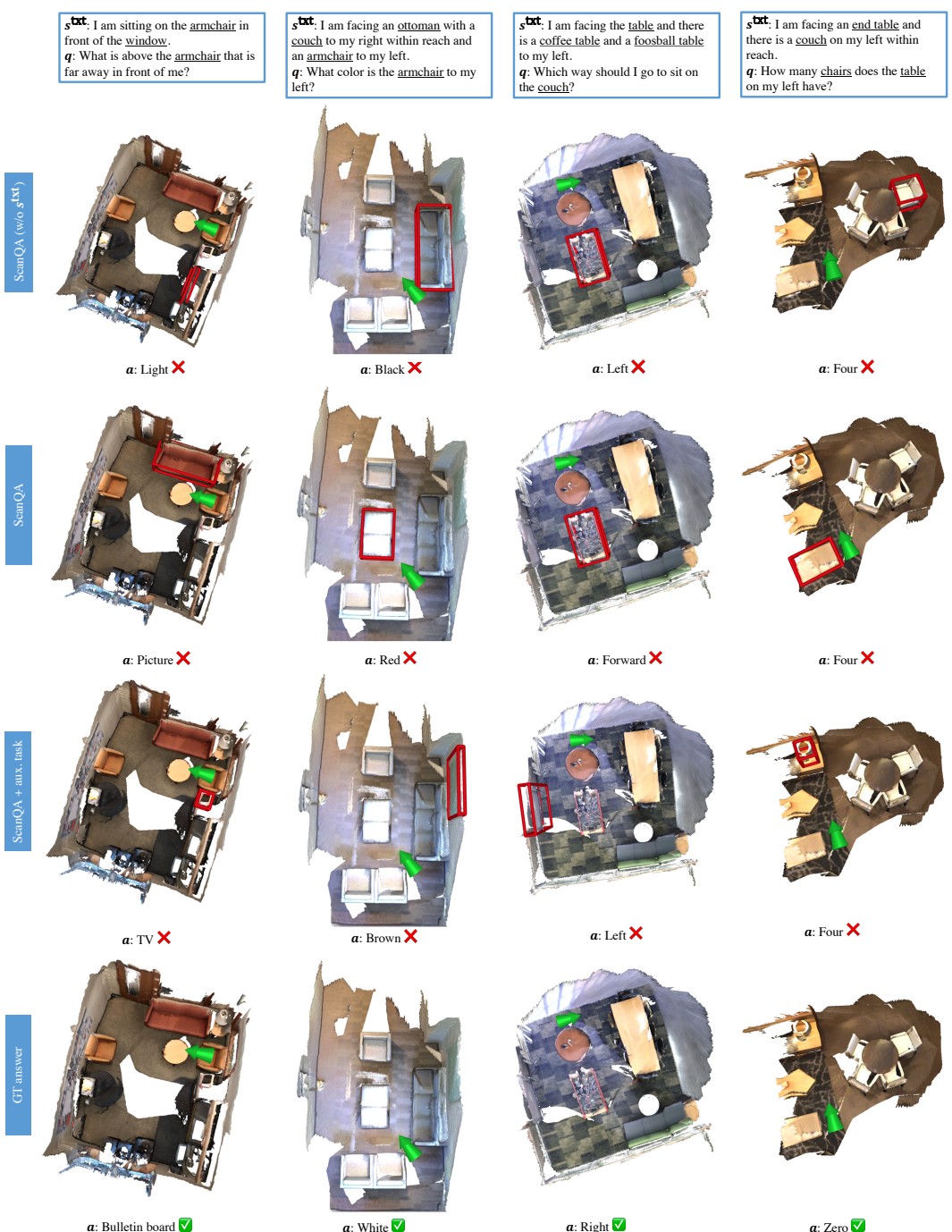

Figure 16: **Failure mode**. Models are likely to predict the wrong answers when they do not attend to relevant objects.

Table 4: Hyper-parameters for the considered models.

| Parameter | Value |
|---|---|
| *ScanQA(modified)* | |
|     Optimizer | Adam |
|     Gradient clipping norm | 1.0 |
|     Epsilon | $1e^{-8}$ |
|     Weight decay factor | $1e^{-5}$ |
|     Beta hyperparameters for Adam | $[0.9, 0.999]$ |
|     Learning rate | $5e^{-4}$ |
|     Learning rate schedule | No learning rate schedule |
|     Batch size | 16 |
|     Total training epochs | 50 |
|     Number of layers for transformer | 2 |
|     Number of heads for transformer | 8 |
|     MLP hidden size in MCAN | 256 |
|     MCAN flatten output size | 512 |
|     Model hidden size | 256 |
|     Number of VoteNet output proposals | 256 |
|     Position regression loss weight $\alpha$ for auxiliary task | 1.0 |
|     Rotation regression loss weight $\beta$ for auxiliary task | 1.0 |
| *MCAN* | |
|     Optimizer | AdamW |
|     Gradient clipping norm | 0.5 |
|     Epsilon | $1e^{-8}$ |
|     Weight decay factor | 0 |
|     Beta hyperparameters for Adam | $[0.9, 0.999]$ |
|     Learning rate | $1e^{-4}$ |
|     Learning rate schedule | No learning rate schedule |
|     Batch size | 16 |
|     Total training epochs | 12 |
|     Number of layers for transformer | 6 |
|     Number of heads for transformer | 8 |
|     MLP hidden size in MCAN | 512 |
|     MCAN flatten output size | 1024 |
|     Model hidden size | 512 |
| *ClipBERT* | |
|     Optimizer | AdamW |
|     Gradient clipping norm | 5.0 |
|     Epsilon | $1e^{-6}$ |
|     Weight decay factor | $1e^{-3}$ |
|     Beta hyperparameters for Adam | $[0.9, 0.98]$ |
|     Learning rate | $5e^{-5}$ |
|     Learning rate schedule | No learning rate schedule |
|     Batch size | 16 |
|     Total training epochs | 10 |

