# OpenReview forum: "SQA3D: Situated Question Answering in 3D Scenes"
_ICLR.cc/2023/Conference — ICLR 2023 poster_

### Official Review · Reviewer_ZAer · 2022-10-25

**Confidence:** 3
**Correctness:** 3
**Technical Novelty And Significance:** 3
**Empirical Novelty And Significance:** 3
**Recommendation:** 6

**Clarity, Quality, Novelty And Reproducibility:**

While the reviewer understands the amount of work and detail that has to be put in for a dataset construction of such scale, section 2 which describes the dataset format, curation and cleaning are quite dense, especially "Multi-staged collection". This makes it difficult for the reader to extract the most important information about the dataset construction.

Some figures could also be improved, e.g., in figures 12-13, the font size is too small. The boxes for the 3D scenes in Figure 7 are too small as well, and it takes quite a lot of zoom-in to understand what each scene contains.


**Strength And Weaknesses:**

Strengths:
- Table 1 presents a clear overview of related work and the contributions of the proposed SQA3D dataset.
- Experiments with 3 model variations, 3D, Video-Image and text-based.

Weaknesses:
- Evaluating GPT-3 on 10% of testing data make results uncomparable. Would be better to replace this with an open-source model, e.g., BLOOM.
- Assuming results reported in Table 3 are top-1 accuracy, the ablation on the important of situation understanding seems to showcase that there is a marginal decrease in most question types when the situation description is removed from the input.
- Please further explain: "No further metric is not included as we find it sufficient enough to measure the differences between baseline models with exact match".


To my understanding, the dataset is built on a standalone web interface rather than on top of an exciting simulator such as AI2-Thor, which could facilitate the evaluation of embodiedQA agents on completing, for example, object navigation tasks. Another comment is on the lack of limitations and societal impact sections. Perhaps articulating similar limitations would make the work more complete, e.g., on the expected usage of the dataset.

**Summary Of The Paper:**

This work proposes a new benchmark for situated embodied scene understanding. Mechanical Turkers are given tasks of diverse scenes, which include writing text descriptions of situations and human activities, writing questions given these tasks and their text descriptions, and finally, answering these questions. The dataset collected spans over 650 rooms and 6 types of questions. Experiments with 3 different model variants and qualitative analysis show there exist a large gap between embodied AI models and humans.

**Summary Of The Review:**

While there exist several benchmark datasets on embodied QA, to the best of my knowledge, this seems to be one of the largest ones and most diverse wrt questions. Concerns involve whether the standalone nature of such datasets and the evaluation with exact match / top-1 accuracy metrics is sufficient for embodied agent scene understanding.

---

### Official Review · Reviewer_cywf · 2022-10-25

**Confidence:** 5
**Correctness:** 3
**Technical Novelty And Significance:** 3
**Empirical Novelty And Significance:** 2
**Recommendation:** 6

**Clarity, Quality, Novelty And Reproducibility:**

Quality：
This article is well-organized. It has completely raised the problems and analysis of 3D scene answering questions, pointed out the key to 3D scene understanding, proposed a new benchmark to reflect scene understanding, and introduced and analyzed the SQA3D dataset in detail, as well as the process of data processing.

Clarity:
The clarity of this article is good, the layout of the article is appropriate, and the figures and tables are useful to describe the data of the proposed dataset, the analysis of the 3D scene understanding model, and the analysis of SQA3D qualitative results enable people to better understand, and the description of the formula in the article is also relatively clear.

Originality:
The paper reflects relatively original contributions. It has proposed a 3D QA benchmark on the basis of ScanNet and proposed a different combination of QA tasks, which has a certain degree of novelty.

**Strength And Weaknesses:**

Strength：
Different from other embedded scene understanding tasks, SQA3D tasks propose to understand and complete tasks in the first person to answer questions, which is more practical and has a broader range of practical task needs. And it has knowledge-intensive questions and a larger scale of the collection. The paper is well-organized, which is easy to read and understand. The current experiment of baseline performance on SQA3D tasks is relatively complete.

Weaknesses：
It is good to introduce how to control the potential biases in the dataset.
The paper should also refer to several recent Situated Reasoning benchmarks or  Video QA benchmarks as they proposed situated reasoning with QA tasks or similar scenarios. For example, STAR: A benchmark for situated reasoning in real-world videos; Agqa: A benchmark for compositional spatio-temporal reasoning, etc.
The experiment in Table 1 shows that the blind model reached 43.65 performance. So the hints or potential connections in the language part are strong which probably will be easy to correlate to answers.
Providing more module-level experiments or ablation studies will be good.

**Summary Of The Paper:**

This article proposes a task to benchmark the understanding of specific agents, scenes, situational questions, and answers in 3D scenes (SQA3D). Given the 3D scene background, then answering a question that requires a lot of situational reasoning from this perspective. Based on scenes from ScanNet, it provides a data set centered on 6.8k unique scenes, 20.4k descriptions, and 33.4k different reasoning questions for these scenes for scenario answers in 3D scenes. At the same time, the author studies the most advanced multi-mode reasoning models, and the results show that these two models still lag behind human performance to a large extent. It shows the key
role of correct 3D representation and the need for better situational understanding in specific scene understanding.

**Summary Of The Review:**

Please refer to the above comments for the pros and cons.

---

### Official Review · Reviewer_4eyF · 2022-10-28

**Confidence:** 3
**Clarity, Quality, Novelty And Reproducibility:** Please see the Strengths and Weaknesses.
**Correctness:** 3
**Technical Novelty And Significance:** 2
**Empirical Novelty And Significance:** 2
**Recommendation:** 8

**Strength And Weaknesses:**

Strengths

- The paper goes into great detail in describing how the dataset SQA3D has been acquired, I appreciate the quality check made in generating the situations.
- The experiments show a thorough analysis of how the methods perform for different question types.
- SQA3D seems to be a useful challenging benchmark where current methods for question answering are far off from human performance opening doors for a lot of advancement opportunities
- I enjoyed reading the detector section as different methods were thoroughly compared in terms of strengths and weaknesses and useful practices like having balanced data were provided.

Weaknesses

-More descriptions need to be given how SQA3D’s challenges differ from existing related benchmarks (Azuma et al., 2022; Ye et al., 2021; Yan et al., 2021). One question to ask, does having a model that performs really well on these existing benchmarks enough to deploy the model, or do we need to test it on SQA3D as well?

- It would be useful to have more detailed descriptions of the dataset challenges and why the scores by the AI algorithms are so slow. One way to reconcile that is to pinpoint several challenges and implement variations of existing methods that can at least slightly address those challenges. This strategy is similar to how some of the challenges of imagenet were addressed using a deep network in its original paper

- No multiple runs of the same experiments in Table 3, it would help to see what  the variance is like if different random seeds were used to run the experiments in order to identify if the differences between the results are significant.

- No code was released to verify and reproduce the results and interact with the dataset to get a deep understanding of how it is structured. Could you add a script that allows us to run one of the methods onto one of the situations? Since this is a benchmark there should be an easy-to-use codebase that makes it smooth to run the baselines and extend them for new algorithms.


**Summary Of The Paper:**

The authors propose a new dataset and benchmark for 3D scene question answering. It is based on ScanNet and it consists of a large number of situations, with descriptions and reasoning questions. These questions test the agent in various ways including relation comprehension, commonsense understanding, and navigation. The authors tested several existing QA methods on this dataset and achieved a very low score in comparison to what humans achieve leaving large room for improvement.

**Summary Of The Review:**

Please see the Strengths and Weaknesses. Overall this work provides a useful benchmark to encourage research in 3D question answering. However, more description is needed for how this benchmark compares to existing ones and it would be useful to have a variation of existing Q&A methods that can at least slightly address this benchmark's specific challenges.

---

### Official Review · Reviewer_neRZ · 2022-10-28

**Confidence:** 4
**Correctness:** 3
**Technical Novelty And Significance:** 2
**Empirical Novelty And Significance:** 4
**Recommendation:** 6

**Clarity, Quality, Novelty And Reproducibility:**

**Clarity**: The ideas are presented clearly and in a structured manner.

**Quality**: The quality of the paper is mostly good, with some minor issues pointed out below which are easily fixable.
- some grammatical mistakes and typos.
- Appropriate use of capitalization in the ‘References’ section.

**Novelty**:
- Low on technical novelty, high on practical novelty and impact as the paper introduces a new task and SOTA baselines on the task in an important area.

**Reproducibility**:
- As noted above, I don’t see authors explicitly stating the release of the code, data and metadata related to the task. If they don’t, then the community may have some difficulty reproducing the results (based on my past experience, authors don’t respond if others’ efforts don’t get the same results and my rating below on reproducibility reflects this.)


**Strength And Weaknesses:**

### Strengths

- The paper addresses creates a novel shared task (challenge) for the emerging field of embodied AI, in particular embodied 3D scene understanding.
- The task is similar to recent work in 3D language grounding and embodies QA but consists of a larger diversity of context-dependent, knowledge-intensive questions on a much larger dataset.
- Human benchmarking is performed.
- Several recent multimodal reasoning approaches are evaluated revealing a large performance gap of ~43%. The choices of these seem reasonable and provide a good coverage of different approaches.
- The large, labeled SQA3D dataset (650 scenes, 20.4k descriptions of ~6.8k situations and 33.4k questions) and the identified performance gap should spur further research on this important topic.

### Weaknesses

- __(W.1)__ I suppose the authors are planning on releasing the code, data and metadata related to the task:  the data cuts used for training their models, the trained models employed to get the results in the paper, the code which changed the SOTA models etc. I might have missed it if the authors explicitly state this. Kindly confirm.

- __(W.2)__ There are some surprising, quantitative results which need further probing and a discussion.

- __(W.2.a) 3D scans__: The VSQ setting should be better than the max of SQ and VQ. This goes to the heart of this task. The improvements seem low on {Is, How, Can} questions. What seems to be going on? It will be good to have human benchmarks in these settings.

- __(W.2.b)  BEV, Ego. videos__: While I agree with the general observations in the paper, low performance on {Is, How, Can, Others} with respect to the blind test is disconcerting. This means that those models are not ‘working’ at all.

 - __(W.2.c) Zero Shot__: Clearly captioning is the bottleneck but for a ‘reasonable’ system, the performance shouldn’t degrade below the ‘Blind test’. The low performance on {Is, How, Can, Others} mirrors the above. Can the authors clarify?


**Summary Of The Paper:**

The paper proposes a novel task, called SQA3D, to benchmark and help improve 3D scene understanding of embodied agents. The embodied agent is provided a visual scene context and a textual description of its own location and state in the scene. The task is to answer an input question which requires both an understanding of the scene and the agent’s location in it. The questions encompass spatial relations, common sense understanding, native and multi-hop reasoning. The authors build the task using scenes from the ScanNet dataset and crowdsourcing with human curation is used to prepare the ground truth. Human benchmark (using amateurs) is prepared on the task yielding an overall score of ~90%. Several state of the art approaches are evaluated with the best reaching 47.2% accuracy. This large gap in performance is expected to facilitate further research on embodied scene understanding.

**Summary Of The Review:**

A good contribution introducing a novel shared task in the general area of embodied AI along with a human study, reasonable baselines based on SOTA identifying the large gap between the two. The paper quality is acceptable. I have some concerns regarding reproducibility and the analysis of the baselines could’ve been better. My lower rating reflects these concerns.

Subject to the above, this task is expected to have a positive impact on the research in the area.

---

### Decision · Program_Chairs · 2023-01-20

**Decision:**

Accept: poster

**Justification For Why Not Higher Score:**

More description is needed for how this benchmark compares to existing ones and it would be useful to have a variation of existing Q&A methods that can at least slightly address this benchmark's specific challenges.



**Justification For Why Not Lower Score:**

A good contribution introducing a novel shared task in the general area of embodied AI along with a human study, reasonable baselines based on SOTA identifying the large gap between the two.

The paper has proposed a 3D QA benchmark on the basis of ScanNet and proposed a different combination of QA tasks, which has a certain degree of novelty.

**Metareview: Summary, Strengths And Weaknesses:**

The paper presents a task, called SQA3D, to benchmark and help improve 3D scene understanding of embodied agents. The embodied agent is provided a visual scene context and a textual description of its own location and state in the scene. The task is to answer an input question which requires both an understanding of the scene and the agent’s location in it. The questions encompass spatial relations, common sense understanding, native and multi-hop reasoning. The authors build the task using scenes from the ScanNet dataset and crowdsourcing with human curation is used to prepare the ground truth.

+ The paper goes into great detail in describing how the dataset SQA3D has been acquired
+ the dataset is a useful challenging benchmark.

- More descriptions need to be given how SQA3D’s challenges differ from existing related benchmarks
- Providing more module-level experiments or ablation studies will be good.

Reviewers reach a consensus this draft is above the acceptance threshold and acceptable. Please refer to reviewers' comment for improving your draft.

**Note From Pc:**

if the above contains the word "oral" or "spotlight" please see: "oral" presentation means -> notable-top-5% and "spotlight" means -> notable-top-25%. As stated in our emails, we are disassociating presentation type from AC recommendations

**Summary Of Ac-Reviewer Meeting:**

N/A